# Diagnostic Reference Levels in pediatric interventional cardiology: A multicenter study by the French cohort in HARMONIC project

Bouchra Habib Geryes[1]*, Soline Bondet De La Bernardie[2], Sébastien Hascoet[3], Charlotte Geant[4], Claire Dauphin[5], Stéphanie Douchin[6], Julia Rousseau[7], François Godart[8], Valérie Pontvianne[9], Caroline Ovaert Reggio[10,11], Bruno Lefort[12], Aurore Danvin[13], Sylvaine Caer-Lorho[2], Marie-Odile Bernier[2], Damien Bonnet[14], Estelle Rage[2]

1 Pediatric Radiology Department, Hôpital Universitaire Necker Enfants Malades, AP-HP, Université Paris Cité, Paris, France, 2 PSE-SANTE/SESANE/LEPID, Autorité de sÛreté nucléaire et de radioprotection (ASNR), Fontenay aux Roses, France, 3 Cardiology department, Hôpital Marie Lannelongue, Le Plessis Robinson, France, 4 Service d'imagerie médicale, Hôpital Marie Lannelongue, Le Plessis Robinson, France, 5 Cardiology Department, CHU Clermont-Ferrand, Clermont-Ferrand, France, 6 Cardiopédiatrie, hôpital couple enfant, CHU Grenoble Alpes, Grenoble, France, 7 CHU Grenoble Alpes, Grenoble, France, 8 Service de Cardiologie Infantile et Congénitale, Institut Cœur Poumon, Lille Cedex, France, 9 Pôle neurosciences et appareil locomoteur, Centre hospitalier universitaire de Lille, Lille Cedex, France, 10 Unité Médicale de Cardiologie Pédiatrique et Congénitale – Timone enfants, AP-HM, Marseille, France, 11 Marseille Medical Genetics – Aix Marseille Université, INSERM U1251, Marseille, France, 12 Institut des Cardiopathies Congénitales, CHRU Tours, Tours, France, 13 PSE-SANTE/SER/UEM, Autorité de sÛreté nucléaire et de radioprotection (ASNR), Fontenay aux Roses, France, 14 M3C-Necker, Hôpital universitaire Necker-Enfants malades, Université de Paris Cité, Paris, France

* bouchra.habib-geryes@aphp.fr

## Abstract

### Background

Pediatric interventional cardiology (IC) is an important diagnostic and therapeutic approach for addressing congenital heart disease (CHD) in children. However, the associated ionizing radiation raises long-term health concerns, emphasizing the need for dose optimization.

### Purpose

To establish multicenter diagnostic reference levels (DRLs) for pediatric IC procedures within the French COCCINELLE cohort as part of the HARMONIC European project, providing data to optimize radiation exposure.

### Methods

A retrospective analysis of radiation dose metrics was conducted on pediatric IC procedures performed across seven hospitals from 2018 to 2020. Data were stratified by five weight groups (<5, 5–<15, 15–<30, 30–<50, 50–<80 kg) and seven IC procedure categories. Median and 75th quantile values for dose-area product (DAP),

**Data availability statement:** All relevant data are within the paper and its Supporting information files. The ethical agreement does not allow data sharing without a data transfer agreement. Any request to access anonymized data should be addressed to our Data Protection Officer (donnees.personnelles@asnr.fr), who will assess the feasibility of sharing the data.

**Funding:** The author(s) received no specific funding for this work.

**Competing interests:** The authors have declared that no competing interests exist.

**Abbreviations:** ASD, Atrial Septal Defect; CHD, Congenital Heart Disease; COD, Coefficient of Dispersion; DAP, Dose-Area Product; DRL, Diagnostic Reference Level; FT, Fluoroscopy Time; HARMONIC, Health Effects of Cardiac Fluoroscopy and Other Imaging in Pediatrics; IC, Interventional Cardiology; ICRP, International Commission on Radiological Protection; IQR, Interquartile Range; PDA, Patent Ductus Arteriosus; RP185, European Guidelines on Diagnostic Reference Levels for Pediatric Imaging; RDMS, Radiation Dose Management System; RDSR, Radiation Dose Structured Report

DAP normalized to body-weight (DAP/BW) and fluoroscopy time (FT) were calculated to define typical values and DRLs, respectively. The DRL-weight curve approach was also used to determine the $50^{th}$ quantile curve for each IC procedure category.

## Results

A total of 1815 pediatric IC procedures were analyzed, comprising 534 diagnostic and 1281 therapeutic procedures. Significant variations in DAP were observed across different procedure types and weight groups, while fluoroscopy time showed no significant variation. Typical-DRL values (median-$75^{th}$ percentile values) of DAP (Gy·cm²) ranged from 1.9–3.9 to 4.8–5.8 for coronarography, from 0.7–1.3 to 9.2–15.6 for angiography, from 0.3–0.6 to 0.8–1.5 for pulmonary valve dilatation, from 2.8–3.7 to 11.6–14.6 for pulmonary artery dilatation and stenting, from 0.6–1.3 to 2.8–6.0 for atrial septal defect closure and from 0.1–0.4 to 3.1–5.4 for patent ductus arteriosus closure in function of weight groups and 0.2–0.3 for Rashkind in patients <5 kg.

## Conclusion

This study provides the first multicenter DRLs for pediatric IC procedures in France, offering valuable benchmarks for dose optimization. These findings lay a strong foundation for future national and international guidelines in pediatric IC care.

## Introduction

Pediatric interventional cardiology (IC) is an essential diagnostic and therapeutic modality for treating congenital heart disease (CHD) in children. However, IC involves exposure to ionizing radiation, which, despite its clinical benefits, raises concerns about long-term cancer risks [1]. In light of these risks, establishing Diagnostic Reference Levels (DRLs) has become a global priority [2–5], serving as benchmarks for radiation doses in medical imaging, and guiding clinical practices towards dose optimization while ensuring interpretable image quality [6–8]. Despite the growing recognition of the importance of radiation safety in pediatric IC, few studies have focused on establishing DRLs for this field. Setting DRLs requires a large number of procedures to ensure statistical robustness, which poses significant challenges in pediatrics due to the relatively low volumes of procedures and the wide variability in patient sizes and clinical indications. Existing studies are mostly limited to single-center studies [9–14], focus on a narrow range of IC procedures that often lack precision in describing the types of procedures covered [15,16], or rely on outdated angiography technology [17–25]. This lack of comprehensive data highlights the need for multicenter studies that encompass a wider array of IC procedures to develop more robust and universally applicable DRLs in pediatric IC.

The ongoing HARMONIC European project [26] aims to develop a European cohort of radiation-exposed children with CHD, to assess the cancer risks associated with ionizing radiation from IC procedures performed during childhood. This

large-scale, multicenter project, funded by the European Commission, seeks to collect detailed data on radiation exposure among pediatric patients with CHD, including medical information on predisposing factors for cancer, quantify organ-specific absorbed radiation doses, and follow patients in the mid and long term to evaluate risks of radiation-associated lympho-hematopoietic malignancies and other cancers. Ultimately, the project aims to provide valuable insights to enhance radiation safety protocols in pediatric IC and guide future diagnostic and therapeutic practices in this vulnerable population. Within the HARMONIC framework, the French COCCINELLE cohort [27–31] includes pediatric patients with CHD who were exposed to radiation through diagnostic and therapeutic IC procedures at multiple French medical centers specialized in pediatric cardiology.

This study aims to address the general lack of comprehensive data on radiation doses in pediatric interventional cardiology. Its primary objective is to establish the first multicenter DRLs in France for various pediatric IC procedure types and body-weight groups. The cohort includes patients ranging from newborns to young adults up to 18 years old, representing a broad spectrum of ages, weights, and CHD types. This multicenter approach enhances the generalizability of the findings and provides valuable insights into radiation practices and outcomes across diverse clinical settings. The newly proposed DRLs can serve as national benchmarks and contribute to filling the data gap in the literature, supporting the development of national and international guidelines in pediatric IC practice.

## Materials and methods

This retrospective study of medical records was conducted in accordance with the ethical principles set forth in the Declaration of Helsinki (1964) and its later amendments, and the requirements of the French and European regulations. All data were fully anonymized before analysis and the data file used for this research was implemented in accordance with French (amended "Informatique et Libertés" law governing data protection) and European General Data Protection Regulation (GDPR) regulations. Each local investigator at every participating center accessed the local data for research purposes between October 1, 2022, and July 17, 2024, in accordance with the protocol approved by the ethical committee, ensuring compliance with ethical standards. The dosimetric data were received by the principal investigator of the COCCINELLE study from October 21, 2022, for the first participating center to July 17, 2024, for the last participating center. The fully anonymized database for the present analysis was constituted on December 13, 2024, by the principal investigator of the COCCINELLE study. This study was conducted retrospectively using anonymized data; therefore, no informed consent was required by the ethics committee. Prior to any study-specific procedure, participants were provided with information regarding the study, in a concise, transparent, intelligible and easily accessible form, in accordance with French (Article L1122-1 of the Code de la Santé Publique) and European (GDPR) low. An information notice was displayed in each participating cardiology department, informing patients and their legal guardians of their rights to access, object to, and restrict the use of their data in this study. The study could not have been carried out on a person if he/she had opposed this, after having received the information mentioned above. This process was approved as part of the ethical approval received from the French national data protection commission. Ethical approval to conduct the study was granted by the French national data protection commission (Commission Nationale de l'Informatique et des Libertés), under approval number 911,112 on December 12, 2011 and deliberation N°2016−1103 on August 11, 2016, amended by deliberation N°2024−1240 on December 30, 2024.

### Study population

As part of the ongoing HARMONIC European project [26], the French national multicenter retrospective cohort, named COCCINELLE [27–31], includes patients with CHD who had not been diagnosed with cancer before the first IC procedure and had undergone at least one first cardiac IC before the age of 16 years old between 2000 and 2020, in 15 participating hospitals specialized in pediatric cardiology. Among them, 7 centers provided data for patients who underwent an IC procedure between January 2018 and December 2020, that were included in this study. This DRL-focused study

concentrated on the most recent three years to ensure that the DRLs reflect the latest equipment technologies. The three-year period also enabled the collection of sufficient statistical data while ensuring that operator practices and equipment characteristics remained consistent across the participating centers. Additionally, recent advancements in data storage systems, such as radiation dose management systems (RDMS), have enabled access to more comprehensive dosimetric data, significantly enhancing the accuracy and reliability of dose analysis. Eligibility criteria included undergoing at least one IC procedure before the age of 18 years. Exclusion criteria included lack of follow-up data and a cancer diagnosis before the first IC procedure. Procedures missing radiation dose records, specifically dose-area product (DAP) data, were excluded from the analysis. Fig 1 illustrates the inclusion and exclusion process for this study. In total, medical records and radiation dose reports were collected for 1712 patients (893 females and 819 males) who underwent 1815 IC procedures included in this cohort.

## Data collection

All angiographic devices underwent annual quality control tests in accordance with national regulations to ensure they were in compliance for use in healthcare settings. Table 1 presents key characteristics of the angiographic devices

Period: January 2000 – December 2020
**26 833 patients**

HARMONIC exclusion criteria:
- First IC > 15 years
- Not followed over time
- Cancer pre first IC
- Incomplete data

**COCCINELLE cohort:**
**23 638 patients**
**31 866 IC procedures**

This study exclusion criteria:
- Age > 18 years
- Period before January 2018
- Not selected IC procedure type or not single procedure
- Missing dose records

**1 712 patients** (893 F, 819 M)
**1 815 IC procedures** (84 coronarography, 450 angiography, 195 pulmonary valve dilatation, 69 Rashkind, 167 pulmonary artery dilatation and stenting, 295 ASD closure and 555 PDA closures)
**Enrolled in this study**

- IC: interventional cardiology; ASD: atrial septal defect, PDA: patent ductus arteriosus, F: female; M: male

**Fig 1. Flowchart of patient selection and enrollment.**

**Table 1. Characteristics of the angiographic equipment used for pediatric cardiology procedures in the seven centers participating in the COCCINELLE cohort.**

| Center number | Manufacturer* | Model | Installation year | Machine type |
|---|---|---|---|---|
| Center 1 | Siemens | Axiom Artis Zee | 2013 | Biplane |
| Center 2 | Siemens | Axiom Artis Zee | 2007 | Biplane |
| Center 3 | Siemens | Axiom Artis Zee | 2008 | Biplane |
| Center 4 | Philips<br>Siemens | Allura Xper FD10<br>Axiom Artis Zee Floor | 2007<br>2013 | Biplane<br>Monoplane |
| Center 5 | Philips | Allura Xper FD10<br>Allura Xper FD10<br>Azurion Clarity IQ | 2010<br>2015<br>2019 | Biplane |
| Center 6 | Philips | Allura Xper FD10 | 2016 | Biplane |
| Center 7 | Philips | Allura Xper FD10<br>Allura Xper FD20<br>Azurion Clarity IQ | 2009<br>2011<br>2018 | Biplane |

*Philips Medical System, Amsterdam, Netherlands. Siemens Healthineers, Erlangen, Germany.

installed in the participated hospitals. Demographic data and dose-related quantities were automatically extracted from the DICOM Radiation Dose Structured Report (RDSR) using RDMS or were manually obtained from medical records.

## Dose metrics

To establish DRLs, the study used quantities recommended by the International Commission on Radiological Protection (ICRP) [2]. The dose-area product (DAP, Gy·cm²) and fluoroscopy time (FT, minutes) were analyzed. DAP and FT distributions, through median and interquartile range (IQR, 25th and 75th percentiles) values, were analyzed to define the typical values and DRL as recommended by the ICRP 135 [2]. Moreover, for a more comprehensive analysis, DAP normalized to body-weight (DAP/BW) was also evaluated, as it serves as a reliable quantity across pediatric ages [13,21,24].

## Data stratifications

In agreement with ICRP135 [2] and RP185 [32], patients were stratified by body-weight groups: <5 kg, 5–<15 kg, 15–<30 kg, 30–<50 kg, 50–<80 kg. The weight data were missing for only 18 patients (1%) and were approximated using the standard growth curves for children in France [33]. The approximate equivalence of age groups as suggested by RP185 [32] for the purpose of comparing weight-based DRLs with age-based DRLs, is indicated in Table 2.

This study uses a standardized classification system, where each procedure was thoroughly reviewed by a cardiologist to ensure consistent and accurate categorization. Initially developed nationally based on the COCCINELLE study protocol, it was then harmonized at the European level within the Harmonic project to accommodate varying practices across centers. This rigorous approach ensures reliable data, enables meaningful future comparisons, and supports the development of international benchmarks in pediatric IC. Pediatric IC procedures included in this study represent the most frequently performed procedures in the French cohort. Procedures were classified into seven categories of diagnostic and therapeutic IC procedures, as listed in Table 2. Diagnostic procedures included coronarography and angiography. The therapeutic procedures investigated were pulmonary valve dilatation, Rashkind procedure for transposition of the great vessels, pulmonary artery dilatation and stenting, atrial septal defect (ASD) closure and patent ductus arteriosus (PDA) closure.

To ensure the representativeness of the sample for DRL definition, weight and IC procedure sub-groups with less than 20 patients were excluded, by analogy with the methodology proposed in RP185 guidelines [32]. Due to the limited number of participated centers, the proposed DRLs were defined as the third quartile values, while the typical values were

**Table 2. Patient data and dose metrics stratified by IC procedure categories and weight groups.** Approximate equivalence of age group as suggested by RP185 [29] for the purpose of comparing weight-based DRLs with age-based DRLs, is indicated. Data are reported only for bands with at least 20 observations. Bold font is intended as DRL value. Italic font is used for groups not satisfying the sample criteria and are reported for completeness (value not intended as DRL). Normalized DAP to body-weight (DAP/BW) is calculated by IC procedure category. The Coefficient of Dispersion (COD) was also indicated to ease DAP dispersion comparisons of different procedures.

| Interventional cardiology Category | Body weight group | Age group | N | DAP (Gy·cm²) median [IQR] (95%IC) | COD | FT (min) median [IQR] (95%IC) | DAP/BW (Gy·cm²/kg) median [IQR] (95%IC) |
|---|---|---|---|---|---|---|---|
| **Diagnostic procedures** | | | | | | | |
| Coronarography | *<5 kg* | *<1 m* | 13 | *0.5 [0.3-0.6] (0.3-0.6)* | 0.64 | *3.2 [2.4-3.9] (2.4-3.9)* | |
| | *5–<15 kg* | *1 m–<4 y* | 13 | *1.0 [0.8-2.1] (0.8-2.1)* | 1.25 | *2.4 [1.9-7.3] (1.9-7.3)* | |
| | **15–<30 kg** | **4–<10 y** | 28 | **1.9 [1.5-3.9] (1.7-2.6)** | 1.23 | **4.1 [3.0-7.7] (3.3-5.5)** | |
| | **30–<50 kg** | **10–<14 y** | 23 | **4.8 [3.9-5.8] (4.3-5.7)** | 0.39 | **7.2 [3.2-13.7] (3.5-11.6)** | |
| | *50–<80 kg* | *14–<18 y* | 7 | *14.6 [9.3-26.2] (8.2-36.1)* | 1.16 | *6.3 [4.7-7.0] (2.9-13.0)* | |
| | All | All | 84 | 2.5 [0.9-4.9] (1.8-3.5) | 1.56 | 4.1 [2.4-7.3] (3.5-4.9) | 0.13 [0.09-0.17] (0.11-0.15) |
| Angiography | **<5 kg** | **<1 m** | 50 | **0.7 [0.5-1.3] (0.6-1.0)** | 1.25 | **7.8 [4.9-11.7] (6.6-10.2)** | |
| | **5–<15 kg** | **1 m–<4 y** | 223 | **1.6 [0.8-3.1] (1.4-1.8)** | 1.40 | **8.4 [5.3-13.4] (7.4-9.7)** | |
| | **15–<30 kg** | **4–<10 y** | 105 | **3.5 [1.6-5.5] (2.4-4.1)** | 1.10 | **7.8 [5.6-14.0] (7.1-9.4)** | |
| | **30–<50 kg** | **10–<14 y** | 49 | **4.5 [1.5-8.2] (2.3-6.9)** | 1.49 | **8.1 [4.0-14.2] (4.8-10.9)** | |
| | **50–<80 kg** | **14–<18 y** | 23 | **9.2 [3.2-15.6] (5.2-15.3)** | 1.35 | **5.8 [3.3-10.8] (3.3-10.8)** | |
| | All | All | 450 | 1.9 [0.9-4.8] (1.7-2.2) | 2.03 | 8.1 [5.1-13.0] (7.5-8.7) | 0.17 [0.09-0.31] (0.16-0.19) |
| **Therapeutic procedures** | | | | | | | |
| Pulmonary valve dilatation | **<5 kg** | **<1 m** | 95 | **0.3 [0.2-0.6] (0.3-0.4)** | 1.21 | **9.1 [6.8-15.0] (8.3-10.3)** | |
| | **5–<15 kg** | **1 m–<4 y** | 78 | **0.8 [0.4-1.5] (0.6-1.0)** | 1.21 | **7.3 [5.0-11.9] (6.3-10.1)** | |
| | *15–<30 kg* | *4–<10 y* | 13 | *1.5 [1.0-1.6] (1.0-1.6)* | 0.41 | *6.5 [5.3-7.4] (5.1-7.7)* | |
| | *30–<50 kg* | *10–<14 y* | 7 | *4.6 [3.8-5.7] (3.1-5.9)* | 0.42 | *8.3 [6.5-10.7] (5.4-13.0)* | |
| | *50–<80 kg* | *14–<18 y* | 2 | *11.7 [9.6-13.7] (7.5-15.8)* | 0.36 | *14.1 [13.0-15.1] (12.0-16.2)* | |
| | All | All | 195 | 0.6 [0.3-1.3] (0.5-0.7) | 1.81 | 8.4 [6.0-12.8] (7.6-9.4) | 0.10 [0.07-0.19] (0.09-0.12) |
| Rashkind procedure for transposition of the great vessels | **<5 kg** | **<1 m** | 66 | **0.2 [0.1-0.3] (0.1-0.2)** | 1.05 | **3.8 [2.5-6.3] (3.1-5.0)** | |
| | *5–<15 kg* | *1 m–<4 y* | 3 | *1.5 [0.9-3.9] (0.3-6.3)* | 1.97 | *14.1 [10.8-16.8] (7.5-19.6)* | |
| | *15–<30 kg* | *4–<10 y* | | | | | |
| | *30–<50 kg* | *10–<14 y* | | | | | |

**Table 2.** (Continued)

| Interventional cardiology Category | Body weight group | Age group | N | DAP (Gy·cm²) median [IQR] (95%IC) | COD | FT (min) median [IQR] (95%IC) | DAP/BW (Gy·cm²/kg) median [IQR] (95%IC) |
|---|---|---|---|---|---|---|---|
| | 50–<80 kg | 14–<18 y | | | | | |
| | All | All | 69 | 0.2 [0.1-0.3] (0.2-0.2) | 1.20 | 4.3 [2.5-7.3] (3.2-5.1) | 0.06 [0.03-0.10] (0.05-0.07) |
| Pulmonary artery dilatation and stenting | <5 kg | <1 m | 15 | 1.1 [0.4-1.5] (0.5-1.6) | 1.05 | 13.6 [8.1-21.5] (7.9-20.0) | |
| | 5–<15 kg | 1 m–<4 y | 45 | 2.8 [1.4-3.7] (2.1-3.2) | 0.80 | 19.5 [9.9-26.9] (14.5-24.9) | |
| | 15–<30 kg | 4–<10 y | 62 | 5.7 [2.7-8.4] (4.4-6.7) | 0.98 | 16.9 [10.4-25.4] (12.6-21.7) | |
| | 30–<50 kg | 10–<14 y | 26 | 11.6 [7.5-14.6] (7.8-13.9) | 0.60 | 17.5 [12.0-27.3] (12.5-26.6) | |
| | 50–<80 kg | 14–<18 y | 19 | 26.5 [13.7-37.6] (14.0-35.8) | 0.90 | 19.1 [11.3-26.5] (11.3-26.5) | |
| | All | All | 167 | 5.0 [2.2-11.0] (3.9-6.2) | 1.77 | 17.6 [10.2-26.5] (14.3-20.2) | 0.27 [0.16-0.40] (0.24-0.32) |
| Atrial septal defect (ASD) closure | <5 kg | <1 m | 3 | 0.5 [0.3-1.4] (0.2-2.4) | 2.18 | 6.6 [4.8-11.8] (3.0-17.0) | |
| | 5–<15 kg | 1 m–<4 y | 8 | 1.1 [0.5-1.5] (0.5-1.6) | 0.90 | 6.7 [6.0-15.0] (4.2-16.4) | |
| | 15–<30 kg | 4–<10 y | 157 | 0.6 [0.4-1.3] (0.6-0.8) | 1.43 | 4.6 [3.1-7.3] (4.1-5.2) | |
| | 30–<50 kg | 10–<14 y | 94 | 1.8 [0.9-3.1] (1.5-2.3) | 1.18 | 5.7 [3.3-8.7] (4.8-6.6) | |
| | 50–<80 kg | 14–<18 y | 33 | 2.8 [1.5-6.0] (1.9-5.0) | 1.60 | 4.3 [3.0-8.4] (3.4-7.1) | |
| | All | All | 295 | 1.1 [0.6-2.3] (0.9-1.3) | 1.61 | 5.1 [3.1-8.2] (4.5-5.9) | 0.04 [0.02-0.07] (0.03-0.05) |
| Patent ductus arteriosus (PDA) closure | <5 kg | <1 m | 89 | 0.1 [0.1-0.4] (0.1-0.2) | 2.20 | 5.5 [4.2-8.1] (4.8-6.1) | |
| | 5–<15 kg | 1 m–<4 y | 281 | 0.7 [0.4-1.1] (0.6-0.7) | 1.09 | 5.0 [3.4-8.0] (4.5-5.7) | |
| | 15–<30 kg | 4–<10 y | 134 | 1.0 [0.6-1.8] (0.8-1.3) | 1.30 | 5.0 [3.1-7.3] (4.1-5.6) | |
| | 30–<50 kg | 10–<14 y | 41 | 3.1 [1.9-5.4] (2.3-4.4) | 1.13 | 6.0 [3.4-10.3] (4.7-8.6) | |
| | 50–<80 kg | 14–<18 y | 10 | 9.1 [5.9-16.4] (5.7-16.8) | 1.16 | 9.6 [7.6-13.9] (7.4-15.0) | |
| | All | All | 555 | 0.7 [0.3-1.5] (0.6-0.7) | 1.62 | 5.2 [3.6-8.3] (4.9-5.6) | 0.07 [0.04-0.13] (0.06-0.08) |

N = number of patients. 95% IC means 95% Confidence Interval on median values. IQR is interquartile range reported as a measure of variability for DAP and FT median values.

derived using the median values from the entire database containing all the collected data per weight and IC procedure sub-group, as suggested by the ICRP 135 [2]. This approach was deemed advisable as an initial method for defining DRLs in pediatric IC procedures. With a larger sample size in the future, alternative approaches may be considered.

In the absence of national pediatric DRL values, results were compared with published data. Publications reporting DAP median or third quartile values were included. Since several weight grouping methods with different granularity have

been proposed in literature, and, as most of the published DRLs are age-based, comparisons were performed relying on this grouping parameter as well.

## Statistical analysis

Standard descriptive statistics were used to summarize the data. Shapiro-Wilk test was applied to check the normality of the distributions. The Coefficient of Dispersion (COD) was introduced to ease DAP dispersion comparisons of different procedures. Differences between IC procedures as well as between weight groups were tested using the Kruskal-Wallis Multiple-Comparison Z-Value Test (Dunn's Test): statistical significance was set as $p < 0.05$. Spearman's Rho was calculated to investigate correlation between DAP/FT quantities and weight.

Additionally, to address challenges in defining patient groups and the limited available data, the DRL curves proposed by the European Guidelines on Diagnostic Reference Levels for Pediatric Imaging (RP185) [32] and other studies [13,21] were investigated. This approach aimed to mitigate issues related to collecting a sufficient number of procedures from a limited number of centers within a reasonable timeframe.

Regarding the DRL-weight curves, based on DAP in function of the body weight, quantile regression method [13,18,24,34] was applied to find the median curves for each IC procedure category. Pearson's correlation coefficient was used to evaluate the alignment between DRL-weight curve values and collected data. The statistical analysis was performed using R software, version 4.4.0 (R Foundation for Statistical Computing, Vienna, Austria).

## Results

### Collected data

Among the 26,833 patients constituting the French database, 3195 were excluded due to missing follow-up data, a prior cancer diagnosis, incomplete data or a first procedure undergone after 15 years old. These 23,638 patients constitute the French COCCINELLE cohort taking part in the HARMONIC project. Additionally, procedures undergone before 2018 or after 18 years old, not single procedures and procedures with missing DAP data were excluded (Fig 1). This resulted in a total of 1712 patients and 1815 procedures from seven hospitals being included in the analysis, comprising 534 diagnostic and 1281 therapeutic procedures. Philips was the most prevalent manufacturer of angiographic systems (67%), followed by Siemens (33%) (Table 1). All systems were equipped with flat-panel image detectors and were biplane, except for one monoplane system. Four angiographic systems were installed prior to 2010 (33%), with the oldest dating back to 2007. All centers involved in this study are public institutions, as pediatric IC activities in France require specific authorization and are predominantly conducted in public healthcare facilities.

### DRL values

Most of the groups met the criterion previously illustrated, with at least 20 procedures (Table 2). Collected data were not normally distributed, as confirmed by the Shapiro-Wilk test. The median and IQR were calculated for DAP and FT, as presented in Table 2 and Fig 2. The distributions of DAP/BW values are also shown in Fig 3. DRL values for IC procedure categories were established for all body-weight groups, except for <5 kg (coronarography, pulmonary artery dilatation and stenting, ASD closure), 5–<15 kg (coronarography, Rashkind and ASD closure), 15–<30 kg and 30–<50 kg (pulmonary valve dilatation and Rashkind), and 50–<80 kg (coronarography, pulmonary valve dilatation, Rashkind, pulmonary artery dilatation and stenting, and PDA closure), which does not fulfil the sample size criterion. Median-75th percentile values of DAP ranged from 1.9–3.9 to 4.8–5.8 Gy·cm$^2$ for coronarography, from 0.7–1.3 to 9.2–15.6 Gy·cm$^2$ for angiography, from 0.3–0.6 to 0.8–1.5 Gy·cm$^2$ for pulmonary valve dilatation, from 2.8–3.7 to 11.6–14.6 Gy·cm$^2$ for pulmonary artery dilatation and stenting, from 0.6–1.3 to 2.8–6.0 Gy·cm$^2$ for atrial septal defect closure and from 0.1–0.4 to 3.1–5.4 Gy·cm$^2$ for patent ductus arteriosus closure in function of weight groups and 0.2–0.3 Gy·cm$^2$ for Rashkind in patients <5 kg.

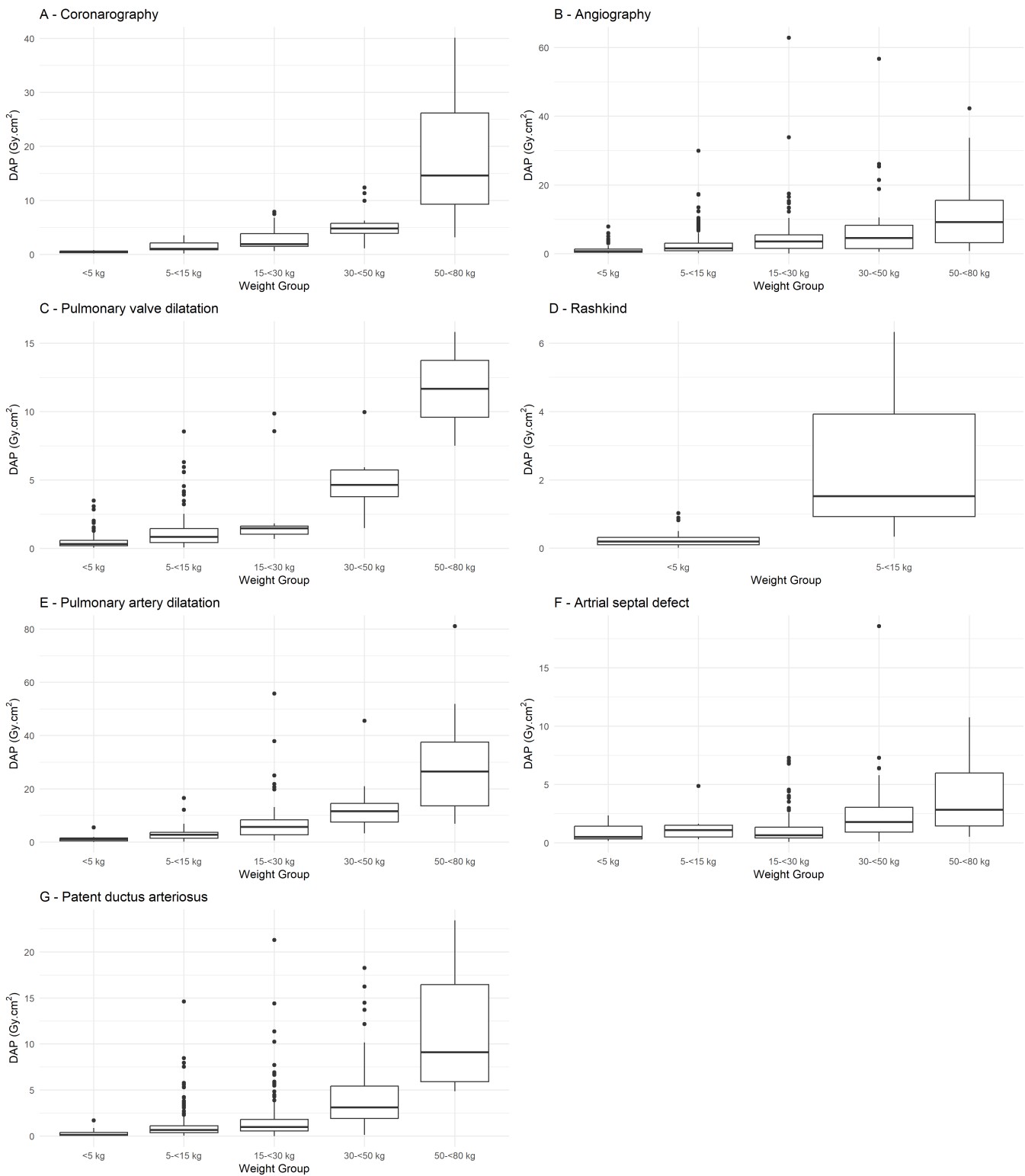

**Fig 2. DAP box plots stratified by body-weight groups for (A) Coronarography, (B) Angiography, (C) Pulmonary valve dilatation, (D) Rashkind, (E) Pulmonary artery dilatation, (F) Atrial septal defect closure, (G) Patent ductus arteriosus closure.** The box represents the interquartile range (IQR), spanning from the 25th percentile (Q1) to the 75th percentile (Q3), with the median (50th percentile) indicated by the line inside the box. Whiskers extend from the box to show the range of the data, capturing values within 1.5 times the IQR from the quartiles. Outliers, defined as values falling outside this range, are marked as individual points beyond the whiskers.

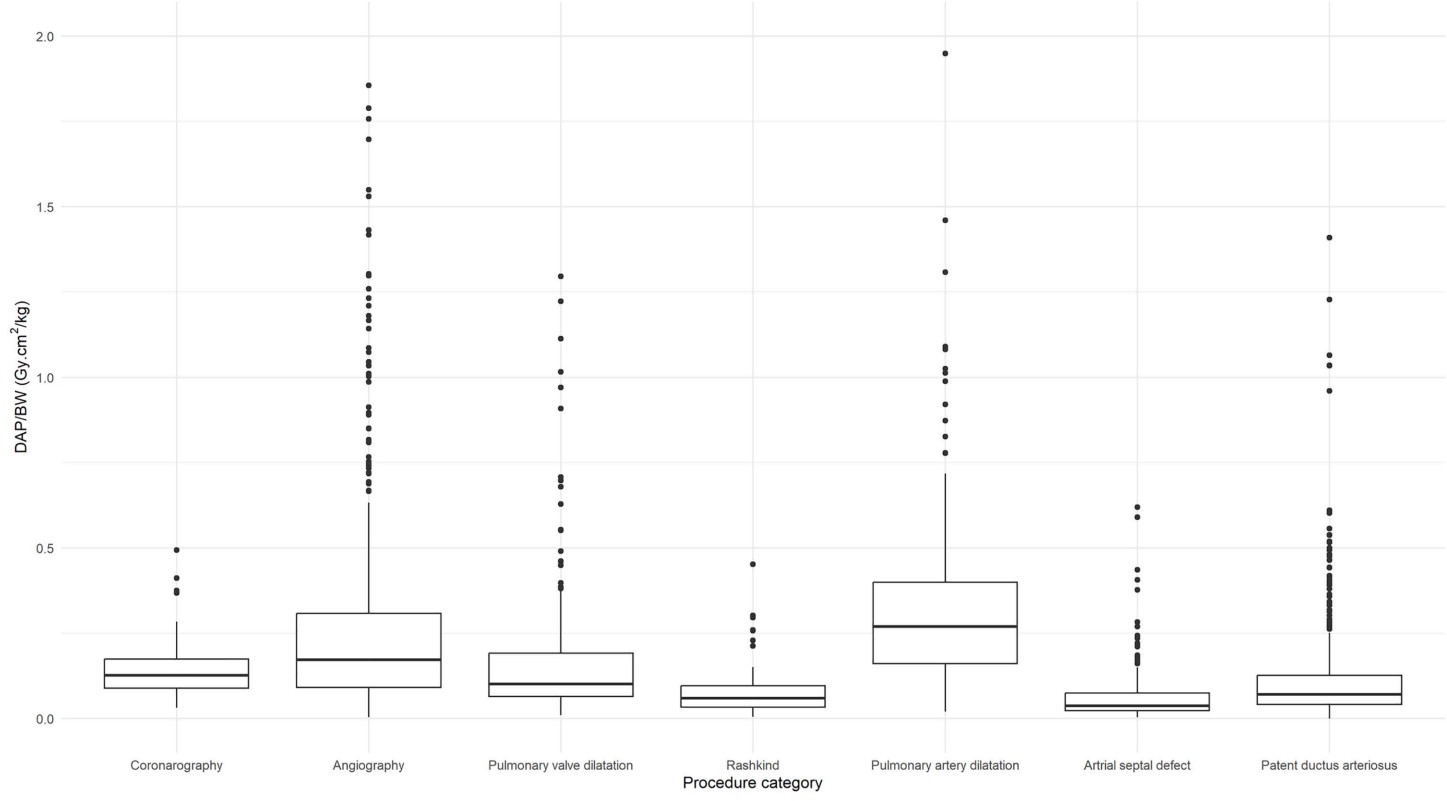

**Fig 3. Box plots of normalized DAP to body-weight (DAP/BW, Gy·cm²/kg) by IC procedure category.**

A wide variability is observed for DAP in each weight group, with the IQR generally being larger for therapeutic procedures. Pulmonary artery dilatation and stenting procedures exhibited the highest values for median DAP, FT and DAP/BW (Table 2, Figs 2 and 3). The results highlight significant variations in DAP across different procedure types and weight groups (Kruskal-Wallis test followed by Dunn's post-hoc test, with p-value < 0.05), whereas no significant variation was observed for fluoroscopy time (p > 0.05). Higher body-weight groups show greater DAP typical values (Table 2 and Fig 2) with a moderate positive correlation between DAP and weight as confirmed by the Spearman coefficients ($\rho = 0.57$), whereas no meaningful correlation was observed between FT and weight ($\rho = -0.0036$).

### DRL curves

The DRL-weight curve approach was used as suggested by other studies [13,21]. Fig 4 illustrates the 50th quantile curve for each IC procedure category. The relationship between DAP and weight is quite well described by a power-law model, as evidenced by the linear trend observed in the log-log scale. The goodness of this fit was further assessed using the Pearson correlation coefficient between collected and calculated DAP values. Table 3 presents the relationship between DAP and body-weight, with the Pearson correlation factor, for all IC procedure categories. The correlation was moderate to strong for all IC procedure categories.

### Discussion

The European DRL Guidelines for Pediatric Imaging [32] highlight that generic DRLs for pediatric imaging are generally insufficient, particularly for therapeutic procedures. These procedures often exhibit higher and more variable radiation

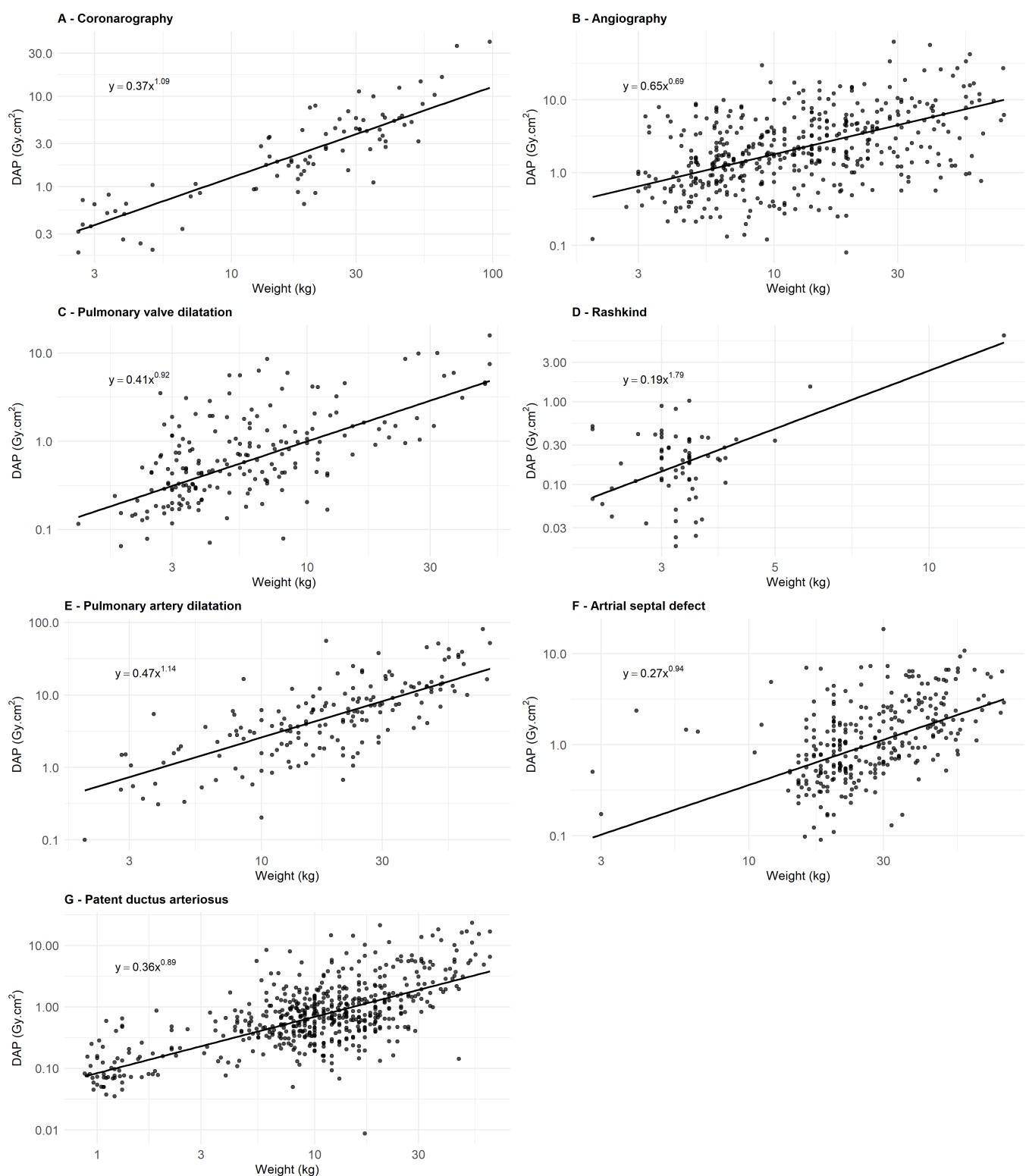

**Fig 4. DAP–weight curves using log-log scales for (A) Coronarography, (B) Angiography, (C) Pulmonary valve dilatation, (D) Rashkind, (E) Pulmonary artery dilatation, (F) Atrial septal defect closure, (G) Patent ductus arteriosus closure.** The 50th quantile curves (black thick line) are shown for each IC procedure category.

**Table 3. Equations, determination coefficients (R²), and Pearson correlation factors (R) for each IC procedure category.**

| Interventional cardiology Category | Equation | R² | R |
|---|---|---|---|
| **Diagnostic procedures** | | | |
| Coronarography | $y = 0.37x^{1.09}$ | 0.75 | 0.87 |
| Angiography | $y = 0.65x^{0.69}$ | 0.22 | 0.47 |
| **Therapeutic procedures** | | | |
| Pulmonary valve dilatation | $y = 0.41x^{0.92}$ | 0.38 | 0.62 |
| Rashkind procedure for transposition of the great vessels | $y = 0.19x^{1.79}$ | 0.18 | 0.42 |
| Pulmonary artery dilatation and stenting | $y = 0.47x^{1.14}$ | 0.56 | 0.75 |
| Atrial septal defect (ASD) closure | $y = 0.27x^{0.94}$ | 0.23 | 0.48 |
| Patent ductus arteriosus (PDA) closure | $y = 0.36x^{0.89}$ | 0.45 | 0.67 |

"y" represents DAP in Gy·cm² and "x" represents the patient weight in kg.

levels. Consequently, current guidelines recommend establishing specific DRLs tailored to pediatric interventions. This study provides a comprehensive analysis of procedure- and weight-specific radiation data for pediatric IC in the French COCCINELLE cohort of patients with CHD, contributing to the HARMONIC European project. It addresses the lack of DRLs for pediatric IC procedures in France by presenting an initial dataset. The results reveal significant variations in DAP across different procedure types and weight groups, underscoring the challenges in implementing DRLs for pediatric IC procedures. Pediatric IC procedures show greater heterogeneity compared to those for adults [32], with radiation doses influenced by patient size, diverse diseases, complexity and variability in procedures, and differences in equipment, staff expertise, and imaging protocols across facilities. This multi-center retrospective study analyzed all procedures performed in seven IC pediatric-specialized hospitals between January 2018 and December 2020 to establish DRLs, in accordance with ICRP 135 [2] and RP 185 [32] recommendations. The findings from this initiative could serve as a basis for future national DRL benchmarks. Institutions exceeding these established levels can identify and implement corrective actions to optimize radiation protocols, such as reducing the image acquisition and fluoroscopy pulse rates, increasing the distance between the radiation source and the patient, applying appropriate collimation, using additional filtration, and removing the anti-scatter grid, and using lower-dose imaging protocols [6–8].

While defining DRLs based on a dataset encompassing all types of therapeutic procedures offers the advantage of larger sample sizes, it certainly introduces bias due to the heterogeneity of procedure types within each weight group and the variations in complexity across different procedures. Indeed, therapeutic procedures showed wider IQRs and consistently higher COD (Table 2), likely reflecting the greater diversity and complexity of therapeutic interventions compared to the relatively standardizable diagnostic procedures. Stratification by procedure type, as recommended in several studies [11,13,18,20,21,24], reduced variability (Table 2) but limited the sample size. Furthermore, the correlation between DAP and weight was moderately positive (Spearman correlation coefficient of 0.57), whereas no correlation was found between FT and weight (Spearman coefficient $\rho = -0.0036$), as also observed in other studies [13,20]. Using this stratified approach, DRL and typical values were established for specific weight groups and IC procedures that met the recommended minimum sample size of 20 patients per subgroup for DRL studies [2,32]: 15–<30 kg and 30–<50 kg for coronarography, <5 kg, 5–<15 kg, 15–<30 kg, 30–<50 kg and 50–<80 kg for angiography, <5 kg and 5–<15 kg for pulmonary valve dilatation, <5 kg for Rashkind, 5–<15 kg, 15–<30 kg and 30–<50 kg for pulmonary artery dilatation and stenting, 15–<30 kg, 30–<50 kg and 50–<80 kg for ASD closure, and <5 kg, 5–<15 kg, 15–<30 kg and 30–<50 kg for PDA closure (Table 2). Tailoring radiological protection measures to account for anatomical differences in pediatric patients remains a priority. This involves refining the normalized DAP/weight ratio to address significant size variations and ensuring minimal radiation exposure without compromising diagnostic quality. This study refined these ratios (Table 2 and Fig 3), proposing

radiation risk categories for procedures and enabling comparisons with similar works. Moreover, this study adopted a DRL-curve methodology [13,21] to address the challenges of pediatric sample characterization. Weight-based DAP-curves, derived using quantile regression, yielded promising results, with the relationship between DAP and weight well described by a power-law model, as evidenced by the linear trend observed in the log-log scale (Fig 4). As illustrated in Fig 2, DAP values varied significantly by procedure category within the same weight group, highlighting the importance of categorizing by procedure type to minimize potential bias when defining DRLs, in line with prior studies [13]. Table 2 and Fig 2 show statistically significant differences in DAP values across weight groups. Therapeutic procedures exhibited higher DAP values compared to diagnostic procedures, likely due to the greater complexity and duration (FT) of therapeutic interventions. For instance, pulmonary artery dilatation and stenting procedures had the highest values for DRL and typical values of DAP, FT and DAP/BW (Table 2). To compare these exposures with the more widely known natural background radiation, the median effective dose from pediatric IC procedures in infants under 5 kg ranges from 0.3 mSv for PDA closure to 3.8 mSv for pulmonary artery dilation and stenting, based on DAP-to-effective dose conversion factors [35]. In comparison, the average annual natural radiation exposure in France is approximately 3 mSv [36].

The lack of standardized methodologies complicates the comparison of DRLs across different studies. Despite numerous publications on pediatric IC, differences in body-weight grouping, IC procedure classifications, and incomplete data hinder direct comparisons. A key strength of this study is the implementation of a standardized classification system, where each procedure was reviewed by a cardiologist to ensure consistent categorization. This classification was first developed at the national level, following the specific protocol of the COCCINELLE study, and then harmonized at the European level in the Harmonic project. In this context, the collection of dosimetric data has been initiated in several participating European countries; however, this work is still in progress, and no national results have been published to date. Median values of dose metrics for stratified procedures in this study were generally within or lower than those reported in previous studies, as summarized in Table 4. The same procedures were selected for comparison with previous studies, but due to the lack of standardization in the naming of procedures, some discrepancies may exist. Comparisons are limited to study groups with at least 20 observations, and the data are affected by substantial variability. Differences among similar surveys may arise from factors such as the technology used (e.g., flat-panel detector systems requiring lower doses for high-quality images), staff training levels, and variations in procedure complexity and protocol optimization. Standardizing methodologies for determining DRLs, in accordance with established recommendations, is essential for enabling reliable comparisons between institutions, harmonizing practices, and optimizing patient exposure.

This study has several limitations. First, DAP values were collected as displayed by the equipment, assuming their accuracy. In France, all equipment undergoes annual quality control checks to ensure that DAP display accuracy remains within a ±25% tolerance. Second, due to the absence of RDMS in several centers, radiation dose data were manually extracted from medical records, which may introduce potential errors. Additionally, as different vendors report radiation doses in varying units, careful conversion was required, highlighting the need for standardization across vendors to reduce such discrepancies. Third, detailed technical parameters of the imaging systems, such as frame rate, filtration, and pulse width, were not systematically collected across all centers and therefore could not be included in the analysis. These parameters typically vary depending on the procedure type as well as the patient's age and weight. However, the DAP inherently reflects the combined influence of these parameters, making it a relevant and robust dose metric in this context. Despite this limitation, the multicenter nature of the study, along with the standardized procedure categorization and weight-based stratification, supports the clinical relevance of the proposed DRLs. While the study focused on DAP and FT to emphasize stochastic radiation effects, incorporating total air kerma and the number of radiographic images could offer additional insights. Moreover, the variability of DAP within the same procedure and weight subgroup may be partly explained by inter-center variability, reflecting differences in clinical practices, equipment, and protocols among centers. This inter-center variability highlights the diversity of real-world clinical settings

**Table 4. Median values of dose-area product DAP (Gy·cm²), fluoroscopy time FT (min) and DAP to body-weight DAP/BW (Gy·cm²/kg) compared with other studies.**

| Interventional cardiology Category | Body weight group | Age group | This study | FT | DAP/BW | Aristizabal et al., 2023 [11] | | De Monte et al., 2020 [13] | | Cevallos et al., 2017 [34] | | Song et al., 2015 [19] | | Ubeda et al., 2015 [18] | | Barnaoui et al., 2014 [20] | | Onnasch et al., 2007 [24] |
|---|---|---|---|---|---|---|---|---|---|---|---|---|---|---|---|---|---|---|
| | | | DAP | | | DAP | FT | DAP | FT | DAP | FT | DAP/BW | FT | DAP | FT | DAP | FT | DAP/BW |
| Coronarography | All | All | 2.5 | 4.1 | 0.13 | 4.5 | 7.2 | | | | | | | | | | | |
| Angiography | All | All | 1.9 | 8.1 | 0.17 | 2.0 | 5.1 | | | | | | | | | | | |
| Pulmonary valve dilatation | <5 kg | <1 m | 0.3 | 9.1 | | | | 0.5 | 9.0 | 2.5 | 19 | | | | | 1 | 5.7 | |
| | 5–<15 kg | 1 m–<4 y | 0.8 | 7.3 | | | | 1.8 | 9.7 | 5.0 | 14 | | | | | 1.4 | 5 | |
| | All | All | 0.6 | 8.4 | 0.10 | 9.7 | 17.6 | | | | | 0.6 | | 6.2 | 11.8 | 1.2 | 6 | |
| Pulmonary artery dilatation and stenting | All | All | 5.0 | 17.6 | 0.27 | | | | | | | | | | | | | |
| Atrial septal defect (ASD) closure | 15–<30 kg | 4–<10 y | 0.6 | 4.6 | | | | | | 6.5 | 17 | | | | | 1.8 | 5.6 | |
| | 30–<50 kg | 10–<14 y | 1.8 | 5.7 | | | | 1.2 | 5.6 | 11.5 | 15 | | | | | 0.7 | 1 | |
| | 50–<80 kg | 14–<18 y | 2.8 | 4.3 | | | | 1.7 | 5.0 | 42.4 | 14 | | | | | 1.1 | 1.5 | |
| | All | All | 1.1 | 5.1 | 0.04 | 9.0 | 6.7 | | | | | 0.37 | 4.9 | | | 0.9 | 1.8 | 0.50 |
| Patent ductus arteriosus (PDA) closure | <5 kg | <1 m | 0.1 | 5.5 | | | | | | 3.2 | 16 | | | | | 2.1 | 6.5 | |
| | 5–<15 kg | 1 m–<4 y | 0.7 | 5.0 | | | | 1.9 | 5.9 | 3.6 | 12 | | | | | 1.4 | 3 | |
| | 15–<30 kg | 4–<10 y | 1.0 | 5.0 | | | | 2.2 | 4.8 | 6.2 | 9 | | | | | 2.8 | 3 | |
| | 30–<50 kg | 10–<14 y | 3.1 | 6.0 | | | | | | 17.4 | 16 | | | | | 4.3 | 3 | |
| | All | All | 0.7 | 5.2 | 0.07 | 7.3 | 7.5 | | | | | 0.36 | 6.5 | 0.8 | 8 | 1.8 | 3 | 0.37 |

captured by our multicenter study, and the proposed DRLs thus reflect a wide range of practices. An analysis of the impact of procedure complexity and operator experience on patient doses was not conducted but would be valuable in future investigations. Lastly, increasing the sample size would enhance the robustness and generalizability of the findings. Extending this work at the European level, using data from multiple centers across several countries as part of the HARMONIC project, will further strengthen the findings and enable broader comparisons and insights across diverse healthcare settings.

## Conclusions

This study establishes the first multicenter DRLs for pediatric interventional cardiology (IC) in France, leveraging French data from the HARMONIC European project. By combining weight-based and procedure-specific stratifications, this study provides a reliable and contemporary benchmark for optimizing radiation dose while ensuring patient safety. These findings mark a significant advancement in radiation safety protocols for pediatric IC procedures and provide a solid foundation for developing future national and international guidelines in pediatric IC care. Future work should focus on periodic updates of these DRLs to reflect advances in imaging technology and evolving clinical practices, ensuring continuous improvement in radiation protection for pediatric patients.

## Acknowledgments

The authors would like to express their gratitude to Jerome Petit, Clement Batteux, Gregoire Albenque, Sarah Cohen and Florence Lecerf for their valuable contributions to data collection.

## Author contributions

**Conceptualization:** Bouchra Habib Geryes, Sébastien Hascoet, Aurore Danvin, Sylvaine Caer-Lorho, Marie-Odile Bernier, Estelle Rage.

**Data curation:** Bouchra Habib Geryes, Sébastien Hascoet, Charlotte Geant, Claire Dauphin, Stéphanie Douchin, Julia Rousseau, François Godart, Valérie Pontvianne, Caroline Ovaert Reggio, Bruno Lefort, Sylvaine Caer-Lorho, Marie-Odile Bernier, Damien Bonnet, Estelle Rage.

**Formal analysis:** Bouchra Habib Geryes, Soline Bondet De La Bernardie, Sylvaine Caer-Lorho, Marie-Odile Bernier, Damien Bonnet, Estelle Rage.

**Investigation:** Bouchra Habib Geryes, Aurore Danvin, Marie-Odile Bernier, Estelle Rage.

**Methodology:** Bouchra Habib Geryes, Aurore Danvin, Marie-Odile Bernier, Damien Bonnet, Estelle Rage.

**Project administration:** Bouchra Habib Geryes, Marie-Odile Bernier, Estelle Rage.

**Resources:** Bouchra Habib Geryes, Sébastien Hascoet, Aurore Danvin, Sylvaine Caer-Lorho, Marie-Odile Bernier, Damien Bonnet, Estelle Rage.

**Software:** Bouchra Habib Geryes.

**Supervision:** Bouchra Habib Geryes, Marie-Odile Bernier, Estelle Rage.

**Validation:** Bouchra Habib Geryes, Sébastien Hascoet, Marie-Odile Bernier, Damien Bonnet, Estelle Rage.

**Visualization:** Bouchra Habib Geryes, Marie-Odile Bernier, Estelle Rage.

**Writing – original draft:** Bouchra Habib Geryes.

**Writing – review & editing:** Bouchra Habib Geryes, Sébastien Hascoet, Charlotte Geant, Claire Dauphin, Stéphanie Douchin, Julia Rousseau, François Godart, Valérie Pontvianne, Caroline Ovaert Reggio, Bruno Lefort, Aurore Danvin, Sylvaine Caer-Lorho, Marie-Odile Bernier, Damien Bonnet, Estelle Rage.

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
