## [Decision Letter · Decision Letter 0]

15 May 2025

Dear Dr. HABIB GERYES,

Thank you for submitting your manuscript to PLOS ONE. After careful consideration, we feel that it has merit but does not fully meet PLOS ONE’s publication criteria as it currently stands. Therefore, we invite you to submit a revised version of the manuscript that addresses the points raised during the review process.

We look forward to receiving your revised manuscript.

Kind regards,

ARNABJYOTI DEVA SARMA

Academic Editor

PLOS ONE

3. Please include a caption for figure 2A to 2G and figure 4A to 4G.

Additional Editor Comments:

Dear Dr. Bouchra HABIB GERYES,

We have received the reports from our advisors on your manuscript, "Diagnostic Reference Levels in pediatric interventional cardiology: a multicenter study by the French cohort in HARMONIC project", which you submitted to PLOS ONE Journal.

Based on the advice received, I have decided that your manuscript could be reconsidered for publication.

Reviewers' comments:

Reviewer's Responses to Questions

**Comments to the Author**

1. Is the manuscript technically sound, and do the data support the conclusions?

Reviewer #1: Yes

Reviewer #2: Yes

Reviewer #3: Yes

2. Has the statistical analysis been performed appropriately and rigorously?

Reviewer #1: Yes

Reviewer #2: Yes

Reviewer #3: Yes

3. Have the authors made all data underlying the findings in their manuscript fully available?

Reviewer #1: Yes

Reviewer #2: Yes

Reviewer #3: Yes

4. Is the manuscript presented in an intelligible fashion and written in standard English?

Reviewer #1: Yes

Reviewer #2: Yes

Reviewer #3: Yes

Reviewer #1: Reviewer -1 Comments on Manuscript PONE-D-25-10735

Title: Diagnostic Reference Levels in Pediatric Interventional Cardiology: A Multicenter Study by the French Cohort in HARMONIC Project

Journal: PLOS ONE

General Assessment:

This manuscript addresses an important topic in pediatric radiation protection by proposing Diagnostic Reference Levels (DRLs) in interventional cardiology based on a large, multicenter French cohort. The topic is of high relevance given the increasing number of pediatric congenital heart disease interventions and the associated long-term risks of radiation exposure.

While the study is generally well-conceived and written, the manuscript will benefit from deeper methodological transparency, stronger discussion of implications, and standardization of terminology and units.

I have made some suggestions as follows:

1. Abstract:

1. The sentence “this study provides…” should be capitalized.

2. Introduction:

1. The introduction would benefit from a clearer definition of study objectives. Is the aim to establish new national DRLs, or compare existing practice to European standards?

2. If there is any prior work done on pediatric DRLs, from other countries (UK, Germany, NCRP), should be cited to situate this work within the broader literature.

3. Materials and Methods:

1. There is insufficient detail on the imaging systems used (e.g., frame rates, filtration, pulse width).

2. The method for constructing the DRL-weight curve is not adequately explained. Please describe the mathematical approach e.g., regression model type.

3. Can you please specify the weight groupings and their justification. Were these based on existing DRL recommendations e.g., ICRP 135?

4. Results:

1. Rationale for no significant variation in fluoroscopy time (FT) across procedures is unclear. This contradicts typical expectations.

2. Is it possible to increase the resolution of figures/graphs you have uploaded?

3. Ranges of DAP and FT should always be accompanied by units and clear specification e.g., Median DAP = 2.5 Gy·cm², 75th percentile = 3.9 Gy·cm².

5. Discussion:

1. Adding more literature in discussion, how do the proposed DRLs compare with those from the UK, Germany, or ICRP will make this part more valuable.

2. Address potential limitations related to inter-center variability in dose protocols, fluoroscopy time settings, operator experience, and patient complexity.

6. Conclusion:

1. A recommendation for future work (e.g., periodic updates, inclusion of newer technology) would strengthen the conclusion.

7. Ethics and Data Sharing:

1. The ethics approval date December 2024 is after data collection (2018–2020), which raises questions about retrospective authorization. Please clarify.

2. Can you please specify whether anonymized data can be shared upon request.

Reviewer #2: Technical Soundness and Data Support:

The study is technically sound, and the conclusions drawn are well-supported by the experimental data provided. The rationale behind the research is clear, and the findings contribute meaningfully to the growing body of knowledge at the intersection of molecular biology and oncology.

Statistical Analysis:

The statistical analyses appear to have been performed appropriately and rigorously. The choice of statistical tests is suitable for the data type and research design, and the results are reported with sufficient detail and clarity.

Data Availability:

I appreciate that the authors have made all underlying data available. This transparency enhances the credibility of the findings and allows for future replication and further exploration by the research community.

Clarity and Language:

The manuscript is well-written and presented in clear, standard English. The structure is logical and facilitates comprehension, even for readers outside the immediate field of study. Minor editorial polishing could enhance readability further, but overall, the writing is of high quality.

Reviewer #3: PleComment 1 :

In the figure 2(A),2(B),2(C),2(D),2€,2(F),2(G) and 3 the Range,Median Line,Mean and outliers are to be depicted in one corner.

Comment 2:

In the Figure 4(A),4(B),4(C),4(D),4(E),4(F) and 4(G) Linear fit equation to be generated and paste in the figure

ase see the attachment uploaded

**Do you want your identity to be public for this peer review?** For information about this choice, including consent withdrawal, please see our Privacy Policy

Reviewer #1: **Yes: ** Shivam Angiras MSc.

Reviewer #2: No

Reviewer #3: **Yes: ** Jibon Sharma

---

## [Author Response · Author response to Decision Letter 1]

20 May 2025

We express our gratitude for your response and your valuable input. Your feedback, particularly the reviewers’ comments, is greatly appreciated. Thank you very much for your time and valuable insights. Please find in the resubmission letter our point-by-point response to the reviewers’ comments. Based on their suggestions, we submit a revised manuscript in a marked-up copy that highlights changes made to the original version.

Kindest regards, yours sincerely,

---

## [Decision Letter · Decision Letter 1]

28 Aug 2025

Diagnostic Reference Levels in pediatric interventional cardiology: A multicenter study by the French cohort in HARMONIC project

PONE-D-25-10735R1

Dear Dr. HABIB GERYES,

We’re pleased to inform you that your manuscript has been judged scientifically suitable for publication and will be formally accepted for publication once it meets all outstanding technical requirements.

Kind regards,

Giuseppe Andò, M.D., Ph.D.

Academic Editor

PLOS ONE

Additional Editor Comments (optional):

Reviewers' comments:

Reviewer's Responses to Questions

**Comments to the Author**

Reviewer #1: All comments have been addressed

Reviewer #2: All comments have been addressed

2. Is the manuscript technically sound, and do the data support the conclusions?

Reviewer #1: Yes

Reviewer #2: Yes

3. Has the statistical analysis been performed appropriately and rigorously?

Reviewer #1: Yes

Reviewer #2: Yes

4. Have the authors made all data underlying the findings in their manuscript fully available?

Reviewer #1: Yes

Reviewer #2: Yes

5. Is the manuscript presented in an intelligible fashion and written in standard English?

Reviewer #1: Yes

Reviewer #2: Yes

Reviewer #1: Accept the current version. All comments have been addressed properly. Manuscript is acceptable to get published.

Reviewer #2: This is a well-executed and timely multicenter study addressing a critical aspect of pediatric interventional cardiology. The establishment of diagnostic reference levels (DRLs) based on a robust dataset provides valuable benchmarks for radiation dose optimization. The methodology is sound, and the stratification by weight and procedure type enhances clinical relevance. This work significantly contributes to improving safety standards in pediatric care and will be instrumental in shaping future national and international guidelines.

**Do you want your identity to be public for this peer review?** For information about this choice, including consent withdrawal, please see our Privacy Policy

Reviewer #1: **Yes: ** Shivam Angiras

Reviewer #2: **Yes: ** Bimugdha Goswami

---

## [Editor Report · Acceptance letter]

PONE-D-25-10735R1

PLOS ONE

Dear Dr. Habib Geryes,

I'm pleased to inform you that your manuscript has been deemed suitable for publication in PLOS ONE. Congratulations! Your manuscript is now being handed over to our production team.

Kind regards,

on behalf of

Prof. Giuseppe Andò

Academic Editor

PLOS ONE